# Efficacy of Toluidine Blue—Mediated Antimicrobial Photodynamic Therapy on *Candida* spp. A Systematic Review

**DOI:** 10.3390/antibiotics10040349

**Published:** 2021-03-25

**Authors:** Rafał Wiench, Dariusz Skaba, Jacek Matys, Kinga Grzech-Leśniak

**Affiliations:** 1Department of Periodontal Diseases and Oral Mucosa Diseases, Faculty of Medical Sciences in Zabrze, Medical University of Silesia, 40-055 Katowice, Poland; rwiench@sum.edu.pl (R.W.); dskaba@sum.edu.pl (D.S.); 2Laser Laboratory Dental Surgery Department, Medical University of Wroclaw, 50-425 Wroclaw, Poland; jacek.matys@wp.pl; 3Department of Periodontics, School of Dentistry, Virginia Commonwealth University, Richmond, VA 23284, USA

**Keywords:** aPDT, *Candida*, diode laser, oral candidiasis, oral microbiome, planktonic cells

## Abstract

The effectiveness of antimicrobial photodynamic therapy (aPDT) in the treatment of oral yeast infections was examined many times in recent years. The authors of this review tried to address the question: “Should TBO (toluidine blue ortho)-mediated aPDT be considered a possible alternative treatment for oral candidiasis?”. PubMed/Medline and the Cochrane Central Register of Controlled Trials (CEN-TRAL) databases were searched from 1997 up to the 27th of October 2020 using a combination of the following keywords: (Candida OR Candidiasis oral OR Candidosis oral OR denture stomatitis) AND (toluidine blue OR photodynamic therapy OR aPDT OR photodynamic antimicrobial chemotherapy OR PACT OR photodynamic inactivation OR PDI). Animal studies or in vitro studies involving *Candida albicans* (*C. albicans*) and/or nonalbicans stain, randomized clinical trials (RCT) involving patients with oral candidiasis or denture stomatitis published solely in English language were included. *Candida* elimination method in animal, in vitro studies and RCT used was TBO-mediated aPDT. Exactly 393 studies were taken into consideration. Then, after analyzing titles and abstracts of said studies, 361 were excluded. Only 32 studies ended up being selected for in-depth screening, after which 21 of them were included in this study. All studies reported the antifungal effectiveness of aPDT with TBO against *C. albicans* and non-*albicans Candida*. In studies conducted with planktonic cells, only one study showed eradication of *C. albicans*. All others showed partial elimination and only one of them was not statistically significant. Experiments on yeast biofilms, in all cases, showed partial, statistically significant cell growth inhibition and weight reduction (a reduction in the number of cells—mainly hyphae) and the mass of extracellular polymeric substance (EPS). In vivo aPDT mediated by TBO exhibits antifungal effects against oral *Candida* spp.; however, its clinical effectiveness as a potent therapeutic strategy for oral yeast infections requires further investigation.

## 1. Introduction

*Candida* species (*Candida* spp.) are components of the normal microbiota of the mucosal oral cavity, gastrointestinal system and genitourinary tracts. *C. albicans* is the most prevalent harmless commensal fungus (approximately 60–70% of microorganisms isolated from the oral cavity), followed by other non-*albicans Candida* spp. like *C. glabrata*, *C. parapsilosis*, *C. krusei*, *C. tropicalis* [1,2,3]. In some situations, especially in immunocompromised patients or patients diagnosed with serious underlying diseases, they can cause both superficial (oral candidiasis) or systemic infections. The predisposing factors include, among others: diabetes mellitus, prolonged use of broad-spectrum antibiotics or immunosuppressive drugs, HIV infection, cytotoxic chemotherapies, radiotherapy, xerostomia, tobacco habits, and dental prostheses [4,5,6,7,8]. Particularly unfavorable for the patient is the fact that after the colonization of the oral mucosa and periodontal pockets, *Candida* spp. interacts with other microorganisms of the host’s flora to form a biofilm. Biofilm is an organized, multilayered structure composed of yeast, bacteria (mainly *Streptococcus*), and matrix (glycoproteins and polysaccharides) [9]. This structure provides a type of cover (greater resistance to the host’s defense mechanisms and drugs used in therapy) [3]. It is also a dangerous chronic focus of inflammation (especially in the periodontium) increasing the risk of cerebral strokes, decompensated glycemia, focal and autoimmune diseases [10].

Oral fungal infections are most conventionally handled by making an accurate diagnosis, the identification of the present species, the assessment and elimination of risk factors that are most likely to occur, followed by a treatment consisting of either topical or systemic antifungal agents [2]. The widespread use of these drugs, conduction of too short therapies and/or subliminal doses has resulted in the development of resistance in *Candida* spp. [11,12,13]. This makes it necessary to look for new therapeutic approaches. Alternative pharmacological antifungal agents have been considered, e.g., essential oils (tea tree oil) [14,15], lemongrass oil [16], oregano oil [17,18], *Cinnamomum cassia* [19], colloidal solutions of metal nanoparticles (silver, gold) [20,21], as well as methods such as ozone therapy [22,23,24,25,26,27], photobiomodulation [28,29] and antimicrobial photodynamic therapy (aPDT) [30,31,32,33]. The most promising of those agents seems to aPDT. It is a modern strategy that requires interaction between a photosensitizer (PS) and light of an appropriate wavelength in the presence of oxygen. The mechanism of action of aPDT is shown in Figure 1 and is based on the use of the excitation energy of the PS molecule (TBO) to produce reactive oxygen species or singlet oxygen. The absorption of a light quantum (hv) induces the photosensitizer from the ground singlet state S_0_(GS) to the excited singlet state S_1_(ES). The excited PS molecule can initiate a chain of transformations leading to the production of radical forms of ROS (reactive oxygen species), e.g., superoxide anion 0_2_-•, hydroxyl radical HO_2_• or hydroperoxide radical HO•- type I reaction. If the quantum efficiency of the intersystem crossing (ISC) is significant, long-lived PS molecules are formed in the excited triplet state T1(ES). These, in turn, in their environment can produce excited singlet oxygen ^1^0_2_ from ordinary molecular oxygen ^3^0_2_—a type II reaction [34,35,36,37]. Radicals or singlet oxygen are extremely strong oxidants that contribute to the destruction of the cell membrane (mainly through the oxidation of fatty acids), enzyme inactivation, receptor dysfunction, and DNA damage [34]. Depending on the structure and biochemical composition of the cell, microorganisms show different sensitivity to oxidative processes. The type I reaction occurs mainly in Gram-positive bacteria (the binding of PS to the cell membrane is very fast and the maximum concentration in the membrane is achieved within 1–5 min). A type II reaction occurs in Gram-negative bacteria and yeasts with an additional sealed cell wall. The thin purine channels of the cell wall prevent the passage of PS. Therefore, in yeast aPDT, cationic PS is used, the long contact with the wall reduces their resistance and, consequently, ensures reaching the cell membrane, its damage, and yeast death (necrosis) [34,37]. An example of a photosensitizer operating mainly in the second mechanism is toluidine blue ortho (TBO). Chemically, it is toluidine chloride—the basic metachromatic thiazide dye [38]. Its small molecule, good solubility in water, cationic form, hydrophilic character, and tendency to form dimers facilitate its binding to microorganisms’ cell membranes. A large difference in affinity to the surface of yeast and host cells provides selectivity. Moreover, TBO is tasteless and odorless. It is cheap and nontoxic. Recent studies confirmed the possibility of using aPDT with different PS as an adjunct to conventional therapy in candidiasis [32,39,40,41], dentures stomatitis [42,43], and periodontitis [44,45,46,47]. As a consequence of these nonspecific oxidizing agents, organisms resistant to antifungal and antibacterial agents could be successfully killed by aPDT, and it seems unlikely that they will develop resistance to such therapy [39,48]. The maximum absorption of aqueous TBO solutions depends on its concentration. Therefore, light sources emitting an electro-magnetic wave from a fairly large range of 600–660 nm can be used to excite TBO in a suitably selected concentration [49].

For those reasons, this study was aimed at reviewing the pertinent literature referencing the susceptibility of oral candidiasis to aPDT mediated by TBO.

## 2. Materials and Methods

### 2.1. Focused Question

The issue focused on in this paper was: “Should TBO-mediated aPDT be considered a possible treatment for oral candidiasis?”.

### 2.2. Protocol

In accordance with the PRISMA statement [50], as well as the Cochrane Handbook of Systematic Reviews of Interventions [51], details of the selection criteria are presented in Figure 2.

### 2.3. Eligibility Criteria

The inclusion criteria are comprised of the following:Animal studies involving *Candida albicans* or other non-*albicans* stains;In vitro studies involving *Candida albicans* or other non-*albicans* stains;Randomized clinical trials involving patients with oral candidiasis or denture stomatitis;Candida elimination method used in animal studies, in vitro studies and RCT is TBO-mediated antimicrobial photodynamic therapy.

The reviewers agreed upon the following criteria of exclusion: Studies published in a non–English language;Case reports or serial case;Letters to the editor;Historic reviews;Studies published before 1996;Duplicated publications.

### 2.4. Research Collection Strategy

An electronic search through PubMed/Medline (1997–2020) and the Cochrane Central Register of Controlled Trials (CENTRAL) (1997–2020) databases was conducted. The research was then complemented by a screening of the references of considered articles in an attempt to find any article that was not found during the database search. For purposes of reviewing the available data pertaining to the subject of interest of this study, the following keywords were used: (Candida OR Candidiasis oral OR Candidosis oral OR denture stomatitis) AND (toluidine blue OR photodynamic therapy OR aPDT OR Photodynamic antimicrobial chemotherapy OR PACT OR photodynamic inactivation OR PDI). The citations of all selected full-text studies and relevant reviews were investigated. The only papers considered were the ones with accessible full-text versions. 

### 2.5. Information Sources, Search Strategy, and Study Selection

Two reviewers independently extracted the data that met the inclusion criteria. The data used were as follows: first author, year of publication, title, study design, use of *Candida* spp. in the study, study groups, study results, light source type and parameters, TBO concentration and incubation time. Selected data were enrolled into a standardized Excel spreadsheet.

### 2.6. Assessing Risk of Bias in Individual Studies

In the initial stage of study selection, titles and abstracts of each study were screened independently by the authors to minimize the potential of reviewer bias. The level of agreement among reviewers was determined by the use of the Cohen’s κ test. [52]. Any difference in opinion on the inclusion or exclusion of a study was resolved by discussion between the authors. 

### 2.7. Quality Assessmentand Risk of Bias across Studies

The procedural quality of each study included in the article was evaluated by two reviewers working independently. The criteria used in establishing the study design, implementation, and analysis were based on the presence of key information for:Course of aPDT, i.e.,:Specified photosensitizer concentration (1) or its absence (0);Indicated incubation time (1) or its absence (0);

Given light source parameters as: type, wavelength, power output, fluence and power density (1) or absence of this information (0).

Objectivity and verification of test results:Negative control group (1) or no negative control group (0);Numerical results available (statistics) (1) or its absence (0);Catalogued *Candida* strain/s used in study (1) only noncatalogued strains used (0).

For RCT additionally: at least 10 patients per group, minimum of a 6 month follows up period. The information collected about the studies was graded. Studies were scored on a scale from zero to six points (score 0–2 low, 3–4 moderate, 5–6 high quality of a study), as recommended in the Cochrane Handbook for Systematic Reviews of Interventions [51]. Only the studies scoring 4 points or higher were included in this review. Of the included studies, 19 received a high-quality assessment (15 studies received the maximum score of 6/6 [39,43,53,54,55,56,57,58,59,60,61,62,63,64,65,66]; the remaining 4 studies 5/6 [38,40,42,67]. Only 2 studies received a moderate quality assessment of 4 [59,68] and none of the studies obtained a low-quality assessment. Any disagreements were resolved through discussion until a consensus was reached.

## 3. Results

### 3.1. Study Selection

Originally, 393 studies were identified as subject to analysis. Screening of the abstracts and titles excluded 361 studies. Thirty two studies made it to further full-text analysis, of which 11 were excluded due to not meeting the predefined inclusion criteria. (Table 1) [48,69,70,71,72,73,74,75,76,77,78].

### 3.2. General Characteristics of the Included Studies

Twenty-one studies were included. Table 2 provides an overview of articles that met the inclusion criteria. Nineteen of these studies were in vitro laboratory studies, one was an in vitro study with an additional animal model, and one was exclusively animal-testing-based.

The materials tested in vitro were either planktonic solutions of cells (10 studies) [38,40,53,55,56,59,62,64,65,66,67], biofilm (six studies) [42,43,54,57,61,63], planktonic solutions of cells and biofilm (three studies) [58,60,66], cellulosic coating and seeds (one study) [63], and adhesive patch and seeds (one study) [39]. Animal models included superficially infected skin wounds in mice [40,61], or infection of the oral mucosa in mice after prior pharmacological immunosuppression (pharmacologically induced neutropenia by cyclophosphamide injection) [61].

In vitro studies with planktonic solutions of cells were performed either in 96-well plates [38,40,53,55,56,59,62,64,65,66,67], Eppendorf tubes [64] or tubes [62]. The effectiveness of aPDT on the *Candida* biofilm structures was tested on polystyrene [43], on acrylic resin plates [42,63], or plastic coverslips [61].

The tested *Candida* strains were mainly: *C. albicans* (18 studies), *C. krusei* (3 studies) [42,54,65], *C. glabrata* (three studies) [42,61,65], *C. tropicalis* (two studies) [62,65], *C. parapsilosis* (two studies) [62,65]. These were most often the standard laboratory strains from the American Type Culture Collection (ATCC), but five studies used clinical strains [43,53,58,62,66]. Some of the strains were naturally resistant to azoles [43,53,58,62]. Seventeen studies were carried out only on *Candida* spp. strains and in four studies the following were additionally tested: *Staphylococcus aureus* [38,63,67], *Escherichia coli* [38,64,68], *Clostridium difficile* [68], *Enterococcus faecalis* [64], *Lactobacillus paracasei* [64], *Porphyromonas gingivalis* [64], *Prevotella intermedia* [64], *Aggregatibacter actinomycetemcomitans* [64] *Propionibacterium acnes* [64] and *Pseudomonas aeruginosa* [64].

The experimental groups in the analyzed studies usually included: (L+PS+) submitted to light irradiation in the presence of the photosensitizer; (L+PS−) submitted to light irradiation only; (L−PS+) treated only with a photosensitizer. The negative control group involved leaving the planktonic solution of cells or biofilm in the dark without exposure to light and photosensitizer (L−PS−) for a specified amount of time. In one study, a positive control group was the effect of chlorhexidine solution on the biofilm [43]. 

### 3.3. Characteristics of Light Sources Used in aPDT

The characteristics of the physical parameters of light sources in studies that met the inclusion criteria are presented in Table 3.

In 12 analyzed papers, a light-emitting diode (LED) was used as the source of light: in eight studies they were diodes with a max. emission peak of 630 nm [54,57,58,60,62,64,66,67]; in two studies: 635 nm [39,43]; in one, two types of diodes (631 and 634 nm) were used [67]; and in one: 637 nm [59]. The widest given emission band of these LEDs is ±15 nm [59] and the narrowest is ±5 nm [58,67]. The output power of the LEDs used was in the range of 30–400 mW, and the applied fluences were in the range of 5–200 J/cm^2^.

In five studies, a laser was used to conduct aPDT (in four studies: a diode laser 660 nm [52,53], 650 nm [63] and 635 nm [42]) and in one—the oldest study from 1999—helium neodymium gas laser (He Ne) 632.8 nm [53]. The power output of the lasers used was in the range of 30–400 mW and the applied energy densities of 15.8–39.5 J/cm^2^.

In two studies, lamps with a full spectrum of white light were used with additional filter probes leaving a wavelength of 635 nm [40,61]. In one work a fluorescent lamp was used [68] and in one work a noncoherent light source was used (630 nm) without specifying its type [38].

### 3.4. Characteristics of TBO Used in aPDT

The characteristics of the toluidine blue ortho used in the studies meeting the inclusion criteria are presented in Table 4.

In 12 studies, TBO was the only PS used [39,42,43,53,54,57,58,59,60,62,66,67]. In nine studies, the effectiveness of aPDT with TBO was compared to other photosensitizers, i.e., methylene blue (MB) (five studies) [40,55,56,61,65], new methylene blue (NMBN) (two studies) [40,65], rose bengal (RB) (two studies) [38,68], malachite green (MG) (one study) [56], pL-ce6-poly-L-lysine chlorin (e6) (one study) [38], curcumin (one study) [63], erythrosine (1 study) [63], riboflavin (one study) [64], and S136- novel pentacyclic phenothiazinium photosensitizer (one study) [65]. In four studies, the effectiveness of TBO-mediated aPDT was additionally enhanced with gold nanoparticles [61], chitosan [66,67] and KP-killer decapeptide [64]. In a study by Huang et al., after aPDT therapy, the clinical or ATCC *Candida* strains resistant to azole antifungal agents were additionally treated with fluconazole and posaconazole [58].

The incubation time used in the studies ranged from 30 s to 30 min. However, the most frequently used were 5 min (nine studies) [39,43,53,54,55,56,57,62,63] and 30 min (seven studies) [39,40,58,61,65,66,67]. In studies with the planktonic solutions of *Candida* cells, TBO was mixed with the suspension and left in the dark until irradiation. However, in the case of the studies on biofilms, in three studies [42,43,58] PS was rinsed before irradiation.

## 4. Discussion

All studies that met the inclusion criteria have demonstrated the efficacy of TBO mediated aPDT against *Candida* spp. (efficacy is understood as reduction in the number of cells or reduction in % CFU/mL). In studies conducted with planktonic solutions of cells, only the study of Nielsen et al. showed complete elimination of *C. albicans* cells (LED 635 nm, 37.7 J/cm^2^, 400 mW, TBO concentration 0.226 mM) [64], and all others showed a partial, statistically significant reduction. No significance was noted in one study by Marigo et al. [63]. Additionally, in all cases of experiments conducted on yeast biofilms, a partial inhibition of cell growth and mass reduction was observed after treatment with aPDT and TBO [42,43,54,57,61,63]. This result was not dependent on the phase of biofilm development [57]. There was a noticeable reduction in the number of cells in the biofilm (both in the yeast and hyphae form) [54,57,58]. Moreover, the weight of EPS (extracellular polymeric substance) in the structure of the biofilm decreased by half after treatment with fluence of 50 J/cm^2^ and a TBO concentration of 0.1 mM [58]. The presented changes may have great clinical implications, because the presence of EPS is largely responsible for the drug resistance of the biofilm [58], and cells in the yeast form are more susceptible to aPDT than the filamentous forms [53].

In studies comparing the efficacy of aPDT mediated by TBO on yeast strains, only as lightly better effect was observed in relation to *C. albicans* than other non-*albicans Candida* [42,54,61,62,65]. Additionally, the clinical strains and those that were resistant to azoles showed sensitivity to aPDT [43,53,58,62,66]. It was lower than in the ATCC strains and without resistance, but the addition of0.5 µg/mL of posaconazole or 0.25 µg/mL of fluconazole to the inoculation after 2 h after aPDT (fluence to 50 J/cm^2^, TBO concentration 0.1 mM) resulted in the eradication of *C. albicans* [58]. A single in vitro laboratory experiment [62] and two on animal models [40,61] confirmed the reduced adhesion of *C. albicans* to epithelial cells (even by 34.5%) and their ability to penetrate deep into the epithelium and epidermis. The inhibition of infection by reducing adhesion was further enhanced by the high efficacy of TBO-mediated aPDT carried out in laboratory conditions on skin wounds and on the surface of the mice tongue mucosa [40,61].

The efficacy of photodynamic therapy is strongly dependent on many variable parameters, hence the large discrepancy in the results obtained by the authors. One of the basic factors is the light source. In photodynamic therapy, it is also extremely important that the maximum absorption of the photosensitizer used is aligned with the wavelength emitted. For aqueous TBO solutions, the maximum absorption changes with the concentration of the solution; therefore, when planning the test, the light source should be selected depending on the PS concentration used [49]. This is clearly visible in Table 3 in the column describing the wavelength of the light source used in the experiment, where diode lasers with wavelengths of 660 [55,56], 650 [63], and 635 nm [42], gas laser He-Ne 632.8 nm [53], and LEDs with maximum light emission of 630–635 nm were used. With the use of a fluorescent lamp as in the study by Decraene et al. [68] 16 h of exposure were required for the activation of the photosensitizer in cellulose coating. 

The other important variable element is incubation time (the time between adding PS to the solution and the start of and the concentration of TBO used). Research by Jackson et al. [53] showed that the most appropriate time needed for TBO from the *C. albicans* planktonic solution to wait in the dark was 5 min, and times longer than 10 min resulted in fewer damaged cells. Completely different results were obtained in the study by Donnelly et al. [39]; the most appropriate TBO incubation time in this study was 30 min, and shorter times resulted in lower effectiveness of the therapy; however, they were still statistically significantly different in comparison to the control group. Chien et al. [66] introduced an effective incubation time of just 10 min that remained constant for another 30 and 60 min. It was expressed as the constant number of cells that survived PDT therapy.

The concentration of the photosensitizer used in photodynamic therapy against *Candida* spp. cells is of great importance. Due to the large size of the yeast cell (approx. 25–50 times larger than the bacteria) and more complex structure (thick cell wall consisting of glucans, mannans, chitins and lipoproteins, and the presence of a cell nucleus, separated from the cytoplasm by a nuclear membrane), they require approximately a 10-fold higher concentration of TBO than bacteria [67]. In several studies, determining the optimal concentration of PS was one of the main research goals [39,53,55,57]. The concentrations determined in these studies were, respectively: 2000, 25, 50, and 10 µg/mL. Such large discrepancies can be explained by examining the influence on both planktonic forms of *Candida* spp. and the structure of the biofilm. Due to its complex structure, the concentrations used to destroy the biofilm structures must be 100 times higher [57]. The cytotoxic mechanism of TBO on *Candida* spp. assessed in many studies showed no statistically significant response to *C. albicans* cells [42,53,54,56,59,60,62,65], while it specifically inhibited the growth of *C. krusei* by 20% in concentrations greater than 0.02 µg/mL [54]. Only studies by Barberio et al. [59] and Rosseti et al. [60] showed a reduction in the number of *C. albicans* cells in planktonic solutions. In a study by Rodriques et al. [65] very little cytotoxic effect of TBO on the fibroblastic cells of L929 mice was demonstrated.

Another very important element influencing the efficacy of the photodynamic reaction are the parameters of the settings of the laser, expressed in particular as energy density (fluence). The determination of the most appropriate fluence has been the subject of several studies [38,40,42,54,56,65]. The energy densities were, respectively: 28 J/cm^2^. (biofilm), 30–40 J/cm^2^ (biofilm), 28 J/cm^2^ (planktonic cells), 39.5 J/cm^2^ (planktonic cells), 40 J/cm^2^ (planktonic cells) and 30 J cm^−2^ (planktonic cells). It can be noted that regardless of the type of *Candida* spp. form (biofilm/planktonic cells), there is a repeating range of 30–40 J/cm^2^, where the efficacy of the therapy reaches even 80% [55]. In a study by Garcia et al. [43] the therapy was performed 2× with an interval of 24 h (2× 175.2 J/cm^2^), it resulted in the eradication of *C. albicans*, although the elimination was small after the first irradiation. The effect of the electromagnetic wave alone without the photosensitizer did not affect the reduction of *Candida* cells in almost all evaluated studies, except for the study by Sousa et al. [56], where a statistically significant reduction was noted compared to the control, but much weaker than during the aPDT reaction (diode laser 660 nm, power output 35 mW, energy density 26.3 and 39.5 J/cm^2^). One (but not the main) mechanism of action of TBO-mediated photodynamic therapy causes the formation of ROS that destroys proteins, fats and DNA and, consequently, damages the cell membrane and many cell structures [62,73] of yeast was the subject of research in two experiments [54,60]. The accumulation of ROS was quantified using 2′,7′-dichlorodihydrofluorescein diacetate (H2DCF-DA) staining. The fluorescence intensity of the suspension was measured directly by using Synergy Multi-Detection Microplate Reader with excitation at 485 nm and emission at 530nm.These studies showed an increase in ROS production [54,60] and a positive correlation with an increase in the cell membrane permeability of *C. albicans* (SYTOXgreen, which penetrates only damaged cells, was added) [60]. There was also an increase in macrophage phagocytic activity after the end of aPDT, expressed by an increased amount of endocytosis observed in the microscopic image [60]. Many chemical substances, in the analyzed studies, were tested as additives increasing the efficacy of aPDT mediated by TBO. The most effective ones were: gold nanoparticles, chitosan, and killer decapeptide. Two of them were mixed with a photosensitizer: gold nanoparticles [61] and killer decapeptide [63], and chitosan was added after aPDT [66,67]. Decreasing gold nanoparticles with TBO significantly reduced the survival rate of *C. albicans*, both in the plankton solutions and in the biofilm structure, where hyphae forms dominated in relation to controls, nanogold particles, and PS alone [61]. A similar relationship was observed in the experiment on mice, where the filamentous forms in skin wounds and on the oral mucosa of the tongue were effectively reduced [61]. The addition of 0.25% chitosan immediately after aPDT with TBO at a concentration of 200 µM for 30 min resulted in the complete elimination of *C. albicans* cells in the biofilm structure. Follow-up with confocal microscopy showed fragmentation of the cell wall as well as swelling and irregular shape of the yeast cells [66,67]. The increase in the efficacy of the drugs and antifungal substances administered during or after photodynamic action seems to be the result of damage to the yeast cell wall and better accessibility to the cell membrane. This exposure of the surface also helps the cells of the immune system to function better.

How does TBO compare to other photosensitizers used in aPDT for yeast elimination? Methylene blue [40,55,56,61,65], malachite green [56], rose bengal [38,68], and riboflavin/blue light 460 nm [64] are less effective on yeast, while new methylene blue N [40,65], curcumin [63], erythrosine [63] and chlorin (e6) [38] presented a stronger effect. This put TBO in the middle, but clearly ahead of the older generation dyes. The newly introduced photosensitizers, the effectiveness of which is slightly greater, are the result of medicine’s constant fascination with the developments made possible by the results achieved with substances such as TBO.

## 5. Conclusions

The usage of TBO-mediated aPDT against yeast seems to be a good alternative method of monotherapy in situations in which there is a superficial infection where the etiopathogenic factor are strains that are multi resistant to traditional antimycotics. However, supporting traditional pharmacological methods of treatment with the use of aPDT seems to be a much better solution. Such an approach provides the possibility of increasing the efficacy of yeast destruction many times over. Nevertheless, it is a therapy method that has its clinical limitations consisting of the possibility of treating only localized, superficial, easily accessible areas. Antiyeast aPDT often requires several minutes of incubation time, which is associated either with the need to use adhesive dressings or to completely darken the room in which the treatment is conducted. In clinical applications, a highly necessary activity to perform is the removal of the excess photosensitizer after the incubation time has concluded. This is related to the phenomenon of light being absorbed by excess PS and the creation of the “optical shielding” effect (high superficial absorption and prevention of the penetration of the light into deeper layers of tissues) or the resulting ROS entering into a chemical reaction with PS (photobleaching). aPDT therapy requires the concentration of TBO to be well-matched with the length of the light source used. Despite many laboratory studies, the optimal settings for the physical parameters of light sources have not been established, which translates into a continuous lack of a recommended clinical protocol.

In all available in vivo studies and animal models, TBO-mediated aPDT clearly exhibits antifungal effects against oral *Candida* spp.; however, its clinical efficacy as a potent therapeutic strategy for oral yeast infections requires further investigation.

## Figures and Tables

**Figure 1 antibiotics-10-00349-f001:**
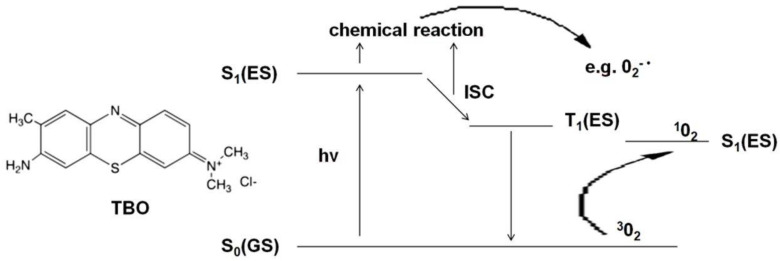
A simplified diagram of Jabłoński showing the mechanism of action of aPDT. Abbreviations: S_0_(GS)—ground singlet state, S_1_(ES)—excited singlet state, hv- light quantum (photon), ICS—intersystem crossing, O_2_^−^•—superoxide anion,^3^0_2_- molecular oxygen, ^1^0_2_- singlet oxygen.

**Figure 2 antibiotics-10-00349-f002:**
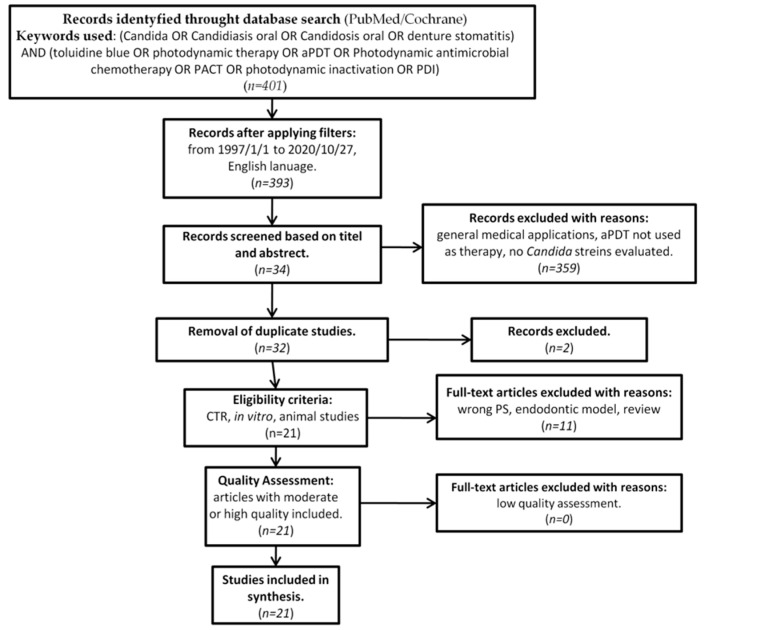
PRISMA flow-chat of selected criteria for the included article reports.

**Table 1 antibiotics-10-00349-t001:** Excluded studies and the reasons for their exclusion.

Ordinal Number	Reason for Exclusion	Reference Number
1	Review	[48]
2	Endodontic model	[69]
3	No aPDT evaluated	[70]
4	No aPDT evaluated	[71]
5	Endodontic model	[72]
6	Endodontic model	[73]
7	Endodontic model	[74]
8	Onychomycosis	[75]
9	Endodontic model	[76]
10	No TBO photosensitizer	[77]
11	Endodontic biofilm model	[78]

**Table 2 antibiotics-10-00349-t002:** Characteristicsof studies that fulfilled the eligibility criteria.

Ordinal Number	Study Design	*Candida* Species	Study Group	Outcomes	Reference Number
1	In vitrostudies 96 well plates	*C. albicans*NCPF 3091 *C. albicans* PHL S3895 *C. albicans* PHLS 8166 planktonic solution of cells	L+PS+ L+PS− L−PS+ L−PS−	*C. albicans* and azole-resistant strains, can be killed by aPDTmediated by TBO. Tested by MTT assay. The best TBO concentration −25 µg/mL.	[51]
2	In vitrostudies 96 well plates	*C. krusei* ATCC 6258 Biofilm	L+PS+ L−PS+ L−PS−	aPDT mediated by TBO reduced *C. krusei* cell growth and biofilm formation. Best TBO concentration 10 µg/mL. Best fluence 40 J/cm^2^.	[52]
3	In vitro studies 96 well plates	*C. albicans* ATCC 10231 Planktonic solution of cells	L+PS+ L−PS−	aPDT mediated by MB and TBO exhibited an antifungal effect against *C.albicans*. pH values in abuffered medium and calcium decreased the inhibition of the yeast growth.	[53]
4	In vitro studies 96 well plates	*C. albicans* ATCC 18804 Planktonic solution of cells	L+PS− L−PS+ L+PS+ L−PS−	aPDT mediated by TBO, MB and malachite green had a fungicidal effect on *C.albicans*. The highest reduction *inC. albicans* was obtained by TBO with energy density of 39.5 J/cm^2^.	[54]
5	In vitro studies 96 well plates	*C. albicans* ATCC 10231 Biofilm	L+PS+ L−PS−	aPDT using TBO exhibited antifungal effects against *C. albicans* biofilm at different stages of development.	[55]
6	In vitro studies 96 well plates	*C. albicans* ATCC MYA-2876D *C. albicans* 2008 no.22 Planktonic solution of cells, biofilm	L+PS+ L−PS− Flu+ L+PS+Flu+ Pos+ L+PS+Pos+	TBO-mediated aPDT could partially remove the extracellular polymeric substance of biofilm. Combination of (aPDT and caspofungin) could kill biofilms and (aPDT and fluconazole or posaconazole) could kill planktonic cells.	[56]
7	In vitrostudies 96 well plates	*C. albicans* Planktonic solution of cells	L+PS+ L+PS− L−PS+ L−PS−	A short time (60 s)TBO-mediated aPDT has a fungicidal effect on *C. albicans*.	[57]
8	In vitrostudies Polystyreneor acrylicresin plates	*C. albicans*SN425 Biofilm	L+PS+ L−PS− CHX+	Twice-daily aPDT on acrylic resinhas reduced *C. albicans* below detection limit, similarly to CHX treatment. After aPDT a pseudohyphae were occasionally visible in biofilm.	[41]
9	In vitro studies 96 well plates	*C. albicans* ATCC 10231 Biofilm, planktonic solution of cells	L+PS+ L−PS−	aPDT using TBO can inhibit both cells growth and biofilm formation by a mechanism evolving the increase in the ROS production, which damages the cell membrane, exposing the nuclear contents.	[58]
10	In vitrostudies Plastic coverslips mice skin wounds infected with C. albicans (BALB/c mice)	*C. albicans*ATCC 90028 *C. glabrata* MTCC 3019 Biofilm	GNPs+ MB+ TBO+ GNPs+MB+ GNPs+TBO+ GNPs+MB+TBO+	The GNPs-PS conjugate combination exhibits synergism in PDT inactivation of *C. albicans* in in vitro and in animal models.	[59]
11	In vitro studies Tubes	*C. albicans*ATCC 18804 *C. albicans* IB05 *C. tropicalis* ATCC 750 *C. tropicalis* CG09 *C. parapsilosis* ATCC 22019 12 clinical strains Planktonic solution of cells	L+PS+ L+PS− L−PS+ L−PS−	aPDT using TBO can have a significant impact on reducing viability or adhesion of *Candida* spp. to buccal epithelial cells including fluconazole resistant *Candida* spp.	[60]
12	In vitro studies Cellulose acetate coating containing the photosensitizer	*C. albicans* Clinical strain	L+PS+ L+PS− L−PS+ L−PS−	The 16 h white light-activated coating with TBO or BR was a simple method of reducing *C. albicans* on surfaces in hospitals.	[61]
13	In vitro studies 96 well plates	*C. albicans* ATCC *18804* Planktonic solution of cells	L+pL-ce6+ L+TBO+ L+RB+	The phototoxicity of TBO toward *C. albicans* was better than RB but much lower than pL-ce6.The highest reduction of *C. albicans* was with energy density of 40 J/cm^2^ and concentration of 50 µg/mL.	[36]
14	In vitro studies Mucoadhesive patch containing TBO and black-walled 96-well microtiter tray	*C. albicans* NCYC *1467* Planktonic solution of cells	L+PS+ L−PS−	aPDT mediated by TBO (30 min. incubation, fluence 200 J/cm^2^ and TBO concentration of 2.0 mg/mL) could total kill of *C. albicans*.	[37]
15	In vitro studies on Sabouraud dextrose agar plates	*C. albicans SC5314* Biofilm	L+PS+KP− L+PS+KP+ L+PS−KP+ L+PS−KP− L−PS+KP+ L−PS−KP+ L−PS+KP−	Combination of red light, TBO or red light, TBO and killer peptides exhibited only partial antifungal effect against *C.albicans* biofilm. Curcumin or erythrosine were significantly better.	[62]
16	In vitro studies on an acrylic resin plates	*C. albicans*ATCC 10231 *C. krusei* ATCC 14243 *C. glabrata* ATCC 15126 Biofilm	L+PS+ L+PS− L−PS+ L−PS−	The efficacy of aPDT against *C. albicans*, *C. glabrata*, and *C. krusei* biofilm has been confirmed. The highest antimycotic efficacy obtained by using laser beam with the parameters of: power 400 mW, fluence 24 J/cm^2^ and time 30 s.	[40]
17	In vitro studies Eppendorf tubes	*C. albicans* ATCC 11775 Planktonic solution of cells	L+PS+ L+PS− L−PS+ L−PS−	aPDT with riboflavin/blue light only resulted in minor reduction sin CFU counts, whereas full kills were achieved for all 8 organisms including *C.albicans* when using TBO/red light.	[68]
18	Animal studies Mice skin wounds infected with *C. albicans*	*C. albicans* CEC749 Planktonic solution of cells	L+MB+ L+TBO+ L+NMB+	PDT with TBO significantly reduced *C. albicans* burden in infected skin abrasion wounds. NMB was superior to MB and TBO in inactivation *C. albicans* in vitro.	[38]
19	In vitro studies 96 well plates	*C. albicans* Planktonic solution of cells	L+PS+ L+PS+Chitosan+	Planktonic cells of *C. albicans* were partially killed by aPDT mediated by 200 µM TBO (fluence 50 J/cm^2^). 0.25% chitosan added for 30 min. after aPDT killed all *C. albicans* cells.	[63]
20	In vitro studies 96 well plates	*C. albicans* ATCC 64548 *C. glabrata* ATCC 90030 *C. krusei* ATCC 6258 *C. parapsilosis* ATCC 22019 *C. tropicalis* ATCC 750 Planktonic solution of cells	L+S136 L+TBO+ L+NMBN+ L+MB+ L+PS−	aPDT mediated by TBO was less effective than NMBN and S136 but more effective than MB.	[64]
21	In vitro studies 96 well plates	*C. albicans* ATCC MYA2876D Clinical strains (2008 no. 19, 22 and 30), planktonic solution of cellsbiofilm	L+PS+ L+PS+Chitosan+	Chitosan augments the killing efficacy on *C. albicans* after aPDT mediated byTBO in planktonic cells and in biofilm.	[65]

Abbreviations: L—laser, PS—photosensitizer, MTT—(3-[4,5-dimethylthiazol-2-yl]-2,5-diphenyl tetrazolium bromide), ATCC—American Type Culture Collection, TBO—toluidine blue, MB—methylene blue, Flu—flucytosine, Pos—posaconazol, RB—rose bengal, pLce6—poly-L-lysine chlorin (e6), GNPs—gold nanoparticles, KP—killer peptide, CHX—chlorhexidine, CFU—colony-forming unit, NMBN—new methylene blue N, S136—novel pentacyclic phenothiazinium photosensitizer, ROS—reactive oxygen species, BALB—Bagg Albino (inbred research mouse strain), NCP—National Collection of Pathogenic Fungi, UK, NCYC—National Collection of Yeast Cultures, Institute of Food Research, UK, MTCC—Microbial Type Culture Collection &Gene BankInstitute of Microbial Technology, India, PHLS—Central Public Health Labolatory Service, UK.

**Table 3 antibiotics-10-00349-t003:** Light sources physical parameters of studies that fulfilled the eligibility criteria.

Ordinal Number	Light Source	Wavelength (nm)	Energy Density (Fluence) (J/cm^2^)	Power Output (mW)	Illumination Time (s)	Spot Size/Fiber Surface Area (cm^2^)	Reference Number
1	He Ne	632.8	21 (J)	35	n.a.	0.03	[51]
2	LED	630	20, 30, 40	68	n.a.	0.38	[52]
3	Diode laser	660	28	30	n.a.	0.38	[53]
4	Diode laser	660	15.8, 26.3, 39.5	35	n.a.	0.38	[54]
5	LED	630	21.7	73	n.a.	0.38	[55]
6	LED	630 ± 5	50	30	n.a.	n.a.	[56]
7	LED	637 ± 15	18	40	60	0.4	[57]
8	LED	635 ± 10	175.2	-	120	n.a.	[41]
9	LED	630	21.47	68	n.a.	0.38	[58]
10	Noncoherent light source	Full spectrum of visible light with filter probes 635	21.6	120	1200	n.a.	[59]
11	LED	630 ± 10	108	100	900	1.5	[60]
12	fluorescent lamp	500–675	-	28,000	216,000	n.a.	[61]
13	Noncoherent light source	630 ± 20	10, 20, 40	n.a.	n.a.	n.a.	[36]
14	LED	635	100, 200	n.a.	n.a.	n.a.	[37]
15	Diode laser	650	10	30	62	0.2	[62]
16	Diode laser	635	12, 18, 24	400, 300, 200	30	0.5	[40]
17	LED	630	37.7	400	60	0.1	[68]
18	Noncoherent light source	Full spectrum of visible light with filter probes 635 ± 15	2, 4, 6, 8, 10	n.a.	n.a.	n.a.	[38]
19	LED	630 ± 5	50	n.a.	1662	n.a.	[63]
20	LED	634 631	5, 10, 15, 25, 30	n.a.	n.a.	n.a.	[64]
21	LED	630	50	30	n.a.	1	[65]

Abbreviations: HeNe—helium neon laser, LED—light emission diode, n.a.—information not available. Fluence: understood as the value of the applied energy in relation to the irradiated surface area. Spot size/Fiber surface area: in contact or in contact no contact mode, understood as the surface area of the used applicator (often calculated from the formula A = pi × r^2^ (r-radius).

**Table 4 antibiotics-10-00349-t004:** Characteristics of TBO used in studies that fulfilled eligibility criteria.

Ordinal Number	Incubation Time (in Minutes)	Concentration/s of PS Used Reference Number	
1	5	3.12, 6.25, 12.5, 25, 50, 100 (µg/mL)	[51]
2	5	5, 10, 20, 50 (µg/mL)	[52]
3	5	10, 50, 100 (µg/mL)	[53]
4	5	100 (µg/mL)	[54]
5	5	10, 20, 50, 100 (µg/mL)	[55]
6	30	0.1 (mM)	[56]
7	n.a.	50 (µg/mL)	[57]
8	5	0.0044 (mM)	[41]
9	10	50, 100 (µg/mL)	[58]
10	30	1 (mM)	[59]
11	5	50, 100 (µg/mL)	[60]
12	n.a.	0.025 (mM)	[61]
13	20	0.05 (mM)	[36]
14	0.5, 3, 5, 30	2000–5000 (µg/mL)	[37]
15	5	0.01 (mM)	[62]
16	1	commercial product with proprietary information about concentration	[40]
17	1	0.266 (mM)	[68]
18	30	20 (mM)	[38]
19	30	2 (mM)	[63]
20	30	2 (mM)	[64]
21	30	20 (mM)	[65]

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
