# Peer review of "Efficacy of Toluidine Blue—Mediated Antimicrobial Photodynamic Therapy on Candida spp. A Systematic Review"

_antibiotics, 2021, doi:10.3390/antibiotics10040349_

Round 1
Reviewer 1 Report
In the manuscript “Efficacy of Toluidine Blue - Mediated Antimicrobial Photodynamic Therapy on Candida spp. A Systematic Review”, Wiench et al. provide an in-depth analysis of the information regarding the efficacy of TBO-mediated antimicrobial therapy on Candida spp. Sytematic reviews are important, especially in the clinical community, since they can be a template for guidelines in the development of therapeutic strategies. The efficacy of antifungals to treat candidiasis is limited; thus, information presented in this study has potential value to pharmacologists and dental communities.
Overall, the research design and interpretation of the experimental findings were solid. However the quality of the paper can be improved by reformatting tables and providing key illustrations in the introduction, methods, and discussion sections. Lastly, while the paper was generally well-written, there were several contextual revisions that should be made prior to accepting the final draft. Specific comments are below.
Contextual Issues
Introduction
A figure showing the chemical reaction that illustrates aPDT activity in respect to TBO would be useful here and provide the reader with mechanistic insight.
Methods section
A figure summarizing the complete protocols (described in sections 2.1-2.8) used in this analysis will be valuable to the reader. This can easily be done by creating a flow chart with Microsoft PowerPoint.
A figure legend should be presented with Table 1. Although lines 130-138 provide a description of the assessment criteria, the individual scoring criteria for each category was not described. A score of 0 or 1 was given. Lastly, table 1 should be reformatted. All words use dashes in the column headers due to the table border sizes.
Discussion
A figure depicting the information described in lines 306 to 326 would be useful.
Typographical and Syntactical issues
- Line 14. quotation mark error and the full name for TBO should be given
- Line 15. Add a question mark after the quoted question.
- Line 23. Extra period should be removed after the word consideration
- Line 30. Italicize In vivo
- Line 49. The reference cited (reference 9) for this statement on Candida biofilms is not appropriate. See Lohse et al’s Nat. Rev. Microbiol., PMID: 29062072, PMCID: PMC5726514 or the authors can use reference 3.
- Lines 62-64 and lines 73-74. These sentences are grammatically awkward and difficult to read. It should be revised for clarity.
- Line 84 quotation mark error
- Lines 93 & 95 (and throughout the manuscript). “in vitro” should be italicized
- Line 106. It is unnecessary to mention the date the search was initiated unless it is standard operating procedure for this type of article.
- Line 199. Two commas follow the word “title”.
- Line 126. A reference should be provided for the Cohen k test.
- Lines 143-145 is sentence fragment. Revise.
- Lines 170. The acronym for the American Type Culture Collection should be ATCC not ACTT
Author Response
The authors would like to thank the Reviewers for giving us the opportunity to revise our manuscript and for useful comments with suggestions. We have responded to reviewers’ concerns and have revised the paper accordingly.
All changes in manuscript are marked in red color.
The following responses have been prepared and detailed corrections are listed below point by point.
Response to Reviewer 1 – Comments
Thank you for revising the manuscript and all the valuable comments that have been proposed. We have made the following changes.
- Line 4 - Co-author would like to change his first name from “Darek” to “Dariusz”.
- Line 14 - Quotation mark has been changed and the full name for TBO was given.
- Line 15 - Question mark was added after the quoted question.
- Line 20&31 - In vivo has been italicized.
- Line 22 - Candida has been italicized.
- Line 23. - Extra period has been removed after the word consideration.
Introduction
As suggested, in the introduction, we have added a description of the mechanism of action of aPDT and a simplified diagram in the form of a figure to clarify the mechanism for the readers.
- Line 30 - In vivo has been italicized.
- Line 39&46 spp. – italic has been removed.
- Line 49. - The change in reference cited (reference 9 has been changed to Lohse MB, Gulati M, Johnson AD, Nobile CJ. Development and regulation of single- and multi-species Candida albicans biofilms. Nat Rev Microbiol. 2018 Jan;16(1):19-31. doi: 10.1038/nrmicro.2017.107. Epub 2017 Oct 3. PMID: 29062072; PMCID: PMC5726514.).
- Lines 62-64 and lines 73-74 - The text has been revised for clarity.
- Fig. 1 has been added with the mechanism of action of TBO-aPDT and with abbreviations.
Materials and Methods
We have added the proposed PRISMA flow-chart to the M&M. At the request of Reviewer 2, because of its illegibility, table 1 was removed and the evaluation quality criteria were described in text.
- Line 84 - Quotation mark has been changed.
- Lines 93 & 95 - In vitro has been italicized.
- Line 106 - The date of the search was deleted.
- Line 199 - One comma following the word “title” has been removed.
- Line 200 - Candida has been italicized. Following abbreviation spp. was added.
- Line 126. - A reference has been provided for the Cohen k test.
- Lines 143-145 were changed.
- Line 170. - The wrong acronym ACTT for the American Type Culture Collection has been changed to ATCC.
References:
- The position [9] was changed to the one proposed by the reviewer.

Reviewer 2 Report
Dear Authors
Your review-manuscript deals with the use of toluidine blue as photosensitizer for antimicrobial PDT and is based on publications of a literature search (PubMed, Cochrane). This interesting review should combine the known underlying scientific background with the clincial experiences. The manuscript showed potential to not only list and summarize the literature but can identify more e.g. unmet needs, unsolved problems, mismatched interpretation, or the limitation of TBO-aPDT.
Although your manuscript seems to be of interest of the community there are some aspects which should be addressed prior to publication. Terms like “we” and “our” should be eliminated as this is a review-manuscript.
Introduction:
It would be of interest to describe the mechanism of the TBO uptake by the microorganism or whether it is attached to the membrane. Furthermore, is there any kind of "selectivity" mechanism especially with regard to surrounding cells? Is the ROS pathway the only pathway of death or is there any 1O2 production? Is the death related to necrosis or apoptosis or others?
Line71: „aPDT may not promote damage to host cells and tissues, therefore, seems to be effective, repeatable and safe to use as an alternative method in oral candidiasis” This sentence induces question like: Why aPDT to not damge host cells and tissues? What is selectivity process? Why is it effective? Are there any log-scale reductions? Why repeatable? Why safe? What is meant by safe? Thus the underlying more mechanism related information are missing which are neccessary to relate these information with the published reports.
L77: „633nm.“ The dot is misspositioned.
It would be helpful to show an absorption spectrum of TBO as shown in (Lucio D’Ilario, Toluidine blue: aggregation properties and structural, aspects Modelling Simul. Mater. Sci. Eng. 14 (2006) 581–595 doi:10.1088/0965-0393/14/4/003) instead of just one specific wavelength. This info also helps about the interpretation (see comments for discussion) of the wavelength and concentration dependent results of the variety of found manuscripts.
M&M
L106: Please add the year 27.10.xyz
L132: Please define pre-irradiation time? Do you mean incubation time?
L133: photosensybilizator -> photosensitizer
L136: Definition of “quality” is missing to understand your sentence. How many papers did you find overall? How many of them reached higher scores as 4? How many paper are excluded due to this defintion?
Tables: Overall the tables can be improved by consecutive and always the same numbering in the first column with just the ref-no in a last coulumn. There is no need of authors name in each of the tables.
Table-1: What is the definition of „given pre-irradiation time“? As there are only 6papers not reaching 6, this table-1 can be withdrawn and the papers can be mentioned in sentences.
L143: „recommended“ either an mistake or s.th. missing.
Results
Table-2: can be shortened
L154: „Table 3„: due to table-3 table-1 is not needed.
Paragraph 3.2. Tables and text should communicate more tricky. In case of numbering the tables one can refer to the consecutive numbering and not by the refs which are not consecutive. The refs are listed in the table and it is not needed to show in the text.
L177: Here the different „pre-irradiation time“ are missing – although may have an impact on the outcome. Which impact can it be?
Table-3: first column as mentioned above.
L189: Regarding the physical parameters of the light sources it should be clarified whether the data were from the refs or self-calculated? If self-calculated – what formula are needed?
Table-4: Column 1 see above. What is the definiton of „application time“? What is meant with „fiber surface area“? Please, explanation of this information „fiber surface area“ is needed. In case filters are used, it would be fine to know the filter characteristics e.g. band width.
L194-206: In this description the abovementioned absorption spectrum, which is pretty broad between 600-650nm, is of interest to argue whether the used wavelengths are appropriate or not. From the physics point of view the cross section of light source emission and excitation by absorption by TBO is of interest to find any relevant differences. Potentially the concentration dependent absorption spectral changes should be taken into account as well.
Table-5: First column see above. Defintion of „pre-irradiation time“ is missing. In case of „-„ one should add whether not available or it is nearby zero. Concentrations should be given in the same SI-unit. Furthermore the way of TBO-application should be mentioned. Furthermore whether the superficial remained TBO is rinsed away from the surface prior to irradiation.
The term „effectiveness“ should be defined and how this term is used in this context.
Discussion:
L227: „effectiveness“: The problem is, that up to now you just mention that there is some action without grading but you did not define any effectiveness or efficay.
L228: what is meant with „inoculation“? perhaps uptake? Please specify.
L230: “… all others showed partial, statistically significant reduction” In turn this would be that the treatment is not effective. But in L227 you said “effectiveness of TBO”, this combination results in confusion. Furthermore in L232 the term “partial inhibition” describes a further effective scenario. Please clarify. Is TBO-aPDT really useful or is there need for improvements or is it useful in some specific applications? Please add critical aspects.
L234: „[52].“ dot is missing
L237: „fluency“ should be changed to „fluence“
2nd paragraph: terms like „effectiveness“, „inoculation“, „fluency“ makes it hard to understand. Please, revise.
L249: “… and their ability to penetrate deep into the epithelium and epidermis” does this mean that C.alb. are not on the surface but more in the depth ? How do they penetrate?
L253-274: First sentence needs explanation, cannot be understood. It is already shown that PDT effects are not related to the coherency of the used light source. Thus the arguments are speculations. A discussion about dosimetry dependency needs to take into account the cross section of the illumination wavelength and the absorption spectrum of TBO plus the potential changes due to the TBO concentration, see abovementioned ref Lucio D’Ilario, and finally the applied irradiance [mW/cm²]. Finally, the photobleaching of TBO needs consideration. Regarding this the irradiation concepts or technologies are of importance and need to be described and compared.
In addition, please clearly differentiate between inoculation and incubation and pre-irradiation time. Furthermore, differentiate what potential uptake-mechanism is the argumentation background in each of the manuscripts. The uptake is often dependent on the applied concentration.
L295: Instead the „fluency“ especially the power density or irradiance in mW/cm² would be of interest as well and should be taken into account and the underlying papers should be investigated regarding such information.
L304-306: What light parameters were used? are there any thermal effects? do the cells may have endogenous PS which may be activated by light?
L306: „formation of ROS“ is not the main mechanism of PDT. There are still 1O2-reactions, please describe and distinguish carefully if you present PDT of TBO-aPDT or … further photosensitzers and their dedicated pathways.
L309: „It was determined by …. fluorescence …“ This sentence is not motviated so far and needs explanation and clarification.
L314-316: Here you mention some sorts of combination therapies. One should mention that PDT and potentially aPDT can induce an increased sensitivity to further therapies, thus broaden the the way to further therapeutic concepts. The question is, whether aPDT results in an increased uptake or permeability of these molecules? Or such combinations may induce further biological effects like immuneresponse and autophagy?
L327: The first sentence needs clarification.
L334: The term „photodynamism“ is not usual and should be exchanged e.g. „light induced non-thermal processes“ or „photodynamic action“
Conclusion:
Here it looks absolute neccessary to distinguish between TBO-aPDT as single treatment or in combination with other therapies. Furthermore, if „effectivness“ is not defined it should be used here. Finally one should find an information whether one should apply TBO-aPDT or not or in what cases it would be beneficial.
Refs:
[24] out of style
Author Response
The authors would like to thank the Reviewers for giving us the opportunity to revise our manuscript and for useful comments with suggestions. We have responded to reviewers’ concerns and have revised the paper accordingly.
All changes in manuscript are marked in red color.
The following responses have been prepared and detailed corrections are listed below point by point.
Response to Reviewer 2 – Comments
review-manuscript.
Thank you very much for your insightful and interesting suggestions for changes. They allowed for a deeper understanding of the topic and a clearer presentation of it to the readers. The following changes were made to the text.
Introduction
- In the introduction, we made major changes to the text regarding the description of the mechanism of action of aPDT (with a figure - at the request of Reviewer 1), the way of interaction with microorganisms (including yeast) and host cells, and the selectivity of the process resulting from them.
Thank you for an interesting additional article, in the introduction, information on the relationship between the concentration of aqueous TBO solutions and the change in the maximum absorption of the substance was included.
M&M
- Figure 2 of the PRISMA flow chart has been added.
- Line 106 - At the request of Reviewer 1, the date of the search has been removed.
- Line 132 – Pre-irradiation time has been changed to incubation time
- Line 133 - the correct word photosensitizer has been implemented
- Table 1 - has been deleted and replaced with chapter text.
- Line 143 - The text has been completed.
Results
- Overall, the tables have been changed in accordance with the recommendations in this article;
- Table2 - has been shortened;
- Line 177 – incubation time - it is analyzed later in this chapter;
- Line 189 – only the spot size (in some articles) was self-calculated. The formula was given below Table 2. Other parameters of the light sources were from references.
- Table 3 – “application time“ has been changed to “illumination time”;
- “Fiber surface area”- has been changed to spot-size – in contact mode = the surface area of the optical fiber.
- Filter probes used with the full spectrum of visible light in positions 13 and 18 (Table 3) parameters related to the width of the LED beam were given and in position 10 – information is not available.
- In Table 5 - ”-“- information not available. The abbreviation has been changed to text for clarity.
- The term „effectiveness“ was wrongly used. In our text, we consider the parameter – efficacy (definition in discussion chapter).
- The information about: the way of TBO-application and whether TBO was rinsed before irradiation? – is given at the end of this chapter.
Discussion
- Line 228 - Inoculation has been changed to the planktonic solution of cells.
- Line 230 – The critical aspects were added in the Conclusion.
- Line 234 – dot has been added.
- Line 237 – wrong word fluency has been changed to fluence.
- 2nd paragraph: terms: „effectiveness“, „inoculation“, „fluency“ have been changed to efficacy, planktonic cells, fluence.
- Line 249 - During the transformation of yeast into hyphae, the microorganisms acquire the features of thigmotropism ensuring growth and infiltration through the pores and intercellular spaces. One of the characteristics of the virulence of a biofilm.
- Line 253-274 - The text has been changed due to its speculative nature.
- Line 295 - The parameter of power density, extremely important in aPDT, was not included in the analysis due to the fact that only a few articles had information about it.
- Line 304-306 – The light source parameters were added but the information on the mechanism of yeast reduction using (L + P-) has not been elucidated. The authors emphasized the surprise and uniqueness of the results. There was no statistical significance.
- Line 306 - The type I reaction mechanism is obviously not dominant in TBO-aPDT and this information has been changed. On the other hand, in the analyzed studies, only ROS was examined without the exact details involved.
- Line 309 - Information on how to quantify ROS, has been completed.
- Line 314-316 – a new text has been added. Probably all the mechanisms described are involved in increasing the sensitivity to other therapies after aPDT. There is too little unambiguous information that is why they have only been mentioned.
- Line 327 - The original sentence has been replaced with questions that allow an answer to evaluate the effectiveness of TBO compared to other PS.
- Line 334 - the incorrect term photodynamism has been replaced by a photodynamic action.
Conclusion
In this part, we have added a text on monotherapy or supporting traditional pharmacotherapy, and information on the limitations of aPDT.
References:
[24] – style has been corrected

Round 2
Reviewer 2 Report
Dear Authors
your manuscript shows great step forward and thus improvements. Thanks for that review.
There are some minor comments to be addressed:
Table 3 and 4: it is usual to mention n.a. in the table as abbreviation for not available.
L169-171 there are two ".-" -> clarify
L228-229 "-" should be changed to ":"
L230: pi should be writte in small font size and normally pi*r²=A is used
L309 and 352: another not neccessary "-"
L334 and 335: a blank is missing between author and "et al."
L359: the closing bracket ")" is missing
Author Response
Dear Reviewer,
of the Special Issue on "Antibiotics Use and Antimicrobial Stewardship" in Antibiotics
The title of our manuscript: Efficacy of Toluidine Blue - mediated antimicrobial Photodynamic Therapy on Candida spp. A systematic review.
Based on the suggestions of the Reviewer, we enclose the latest version of the article with the changes made. Thank you very much on behalf of all the authors for the submitted reviews of our article. We analyzed the whole very carefully and edited. Please find attached all correction in the track changes.